# The New Urban Profession: Entering the Age of Uncertainty

**Rob Roggema [1,2,*] and Robert Chamski [1]**

1    Department of Built Environment, Inholland University of Applied Sciences,
     1817 MN Alkmaar, The Netherlands; robert.chamski@inholland.nl
2    Cittaideale, 6707 LC Wageningen, The Netherlands
*    Correspondence: rob.roggema@inholland.nl

**Abstract:** The context of urbanism is changing rapidly. The context for working in the field of urban design and planning is influenced by the pace of change; uncertainty; and massive transitions. The urban professional, however, is still used to planning for small changes and repeating traditional approaches. In this paper, we have investigated major future tasks and problems that require rethinking the skills required from people working in the urban arena. By conducting in-depth conversation with leading thinkers in the field, the tension between idealism and the urgency to act versus realism and the trust in current systems dominated by economic laws is present. This results in the conclusion that a different skillset is required in order to face future complexities and to be able to connect design creativity with process sensitivity in short- and long-term periods and at small and large scales.

**Keywords:** transformation; urban planning; urban design; urban professional; urgency

## 1. Introduction

Urbanization has, amongst many other factors, an impact on nature and health [1], avian diversity [2] and biodiversity in general [3]; groundwater quality [4]; soil [5]; prices and affordability of housing [6]; energy consumption [7]; land-use change and climate [8]; the regional climate [9]; and urban heat impact [10]. Global urban land expansion [11,12] has significant climate impacts [13]; it influences deforestation [14], biodiversity [15] health, such as increasing obesity [16] and mental health issues [17]. Climate variability and change can exert profound stresses on urban environments, which are sensitive to heat waves, droughts and changes in the frequency and magnitude of flash floods [18]. At the same time, urban areas are centers of wealth, human population and built infrastructure and are, therefore, considered to be 'first responders' to climate change [19]. Moreover, cities are the fundamental units for climate change mitigation and adaptation [20].

Currently, with over 50% of the global population living in cities [21], a number that is expected to rise to 70% in 2050 [22], these impacts deserve attention from urban professionals. Additionally, future problems, such as climate impacts [23,24]; social unrest [25,26] and migration [27–29]; increased inequality [30–32]; and the limitations of natural resources [33–35], i.e., the limits to growth [36–38], potentially accelerate the need for urban responses. Thus far, the urban professional, being an urban planner or designer or an urbanist, is confronted with an enormity of subjects, complexities and uncertainties, which makes it a next to impossible task. The question raised is as follows: what can the urban professional act on in order to deliver a distinguished contribution to creating urban environments that are providing a sustainable, resilient and healthy place for people to live in?

In this article, we explore topics, approaches and competencies of the future urban professional. In Section 2, a brief sketch is provided of some of the professions active in the realm of urban development, including current urgencies that are heading toward them. In Section 3, we describe our methodology. In Section 4, the findings of 25 in-depth

interviews with international and national urban experts are discussed. The article ends with conclusions.

## 2. Urban Planning, Urban Design and Urbanism

In order to understand urban responses to social, ecological or economic developments in the city, the core attributes of professions active in the urban realm need to be comprehended. A "profession" is a special form of community of persons who share the same special type of occupation, whose practitioners assume responsibility for the affairs of others and provide services that are indispensable for the public good [39]. Urban planning, urban design and urbanism each have their own viewpoints on their own profession.

### 2.1. Urban Planning

There is a saying of the following: '(Urban) planning is not politics, but it is in politics' [40]. Politics is primarily concerned with resource distribution, and the role of planning and planners is, inter alia, to provide a reasoned, rational and socially sensitive contribution to political decision making [41]. The nature of planning as a profession comprehends the promotion of general welfare in the public interest, submerging any personal interests to the interests of the client; it is intellectual and varied in character, discrete and just and requires an advanced type of knowledge [42]. The planner fosters awareness; promotes public involvement; and establishes and maintains clear ethical standards [41]. 'Planning with/for people' enhances the 'quality of life' for 'all' in the 'built environment' [43]. Planning must consider not only the interests of the current generation but also the wellbeing of future generations, and it is inherently related to some form of 'common good' [44] (p. 465), which includes cities, towns and their regional surrounding and is aimed at both economic and social goals [45]. Therefore, planning considers the big picture, involves the entire community and looks ahead [46]. However, 'the public image of the planner depicts him as an artist; as the master planner who intuitively prepares the best plans for the community. Planners, through their professional organizations, try to maintain an image of comprehensiveness. Neither explains why the plans prepared by the urban planner fail to gain real social commitment. What urban planners do and why they do it becomes more understandable when they are seen as governmental functionaries.' [47].

> "The master-economist must possess a rare combination of gifts.... He must be mathematician, historian, statesman, philosopher—in some degree. He must understand symbols and speak in words. He must contemplate the particular, in terms of the general, and touch abstract and concrete in the same flight of thought. He must study the present in the light of the past for the purposes of the future. No part of man's nature or his institutions must be entirely outside his regard. He must be purposeful and disinterested in a simultaneous mood, as aloof and incorruptible as an artist, yet sometimes as near to earth as a politician."
>
> —John Maynard Keynes

### 2.2. Urban Design

The following has been said of urban design: 'We know what urban design is not. It is not architecture, not even big architecture. It is not land use policy, sign controls, and street lighting districts. It is also not merely sensitivity to design in the drafting of public policy, nor respect for the urban fabric in which architectural designs are wrapped. We also know that it is not strictly Utopian or procedural, and that it is not necessarily a plan for downtown, however architectonic, nor a subdivision regulation no matter how particular' [48] (p. 67). Urban design is the art of making places in an urban context, which involves designing groups of buildings and the spaces and landscapes between them and to further improve the creation of frameworks for successful development [49]. City beautification was the fundamental purpose of urban design at the time it was introduced as a separate profession. Over time, the scope and objectives of urban design have changed, and urban design currently plays a vital role in city development. Today, urban design

functions at the crossroads of architecture, landscape architecture and city planning. It has become a collaborative discipline functioning with other disciplines to create three-dimensional forms and spaces for people that function effectively [50]. Urban design is the art of three-dimensional city design at a scale larger than a single building, and the urban designer acts as a 'fixer, coordinator, or stimulator rather than a detached observer churning out observations or reports' who is concerned with the total built form, i.e., the production of acceptable townscape. The scope of urban design is primarily the area on the edge of architecture and planning. The urban designer is responsible for the three-dimensional form of the city at the local planning level as both a designer, i.e., a person who engages in a specific creative act that produces a design, and as a controller/negotiator who ensures that an adequate design evolves [51], providing a 'physical design direction to urban growth, conservation, and change...' [52] (p. 12). In addition, quality of life, the public realm and process are seen as significant aspects of the 'thresholds of scale' whereby interrelationships of building site, neighborhoods and districts; the city; metro regions; and 'corridors' are building blocks of design intervention [53]. Therefore, urban design seeks to enhance the life of the city and its inhabitants in socio-economic and environmental terms [54].

### 2.3. Urbanism

The following has been said about urbanism: 'Urbanism is about what happens inside cities, the form and function of cities, and how cities relate to the rural. It often refers to the study of how inhabitants of urban or urbanizing areas interact with the social and built environments of cities. The concept of urbanism is linked to the professions associated with the physical and social design and management of urban structures and communities' [55].

### 2.4. Flaws

Cities have been variously defined and analyzed by population density; geographic size; integrated economies with a diversity of goods and services; the proliferation of specific building types or changes in urban form such as high-rise buildings; high population recreational spaces such as stadiums and theatres; new forms of government and urban governance; or the increasing detachment of a population from directly providing their own food and energy needs [56–62].

Cities might also be defined by what they produce, such as housing wealth or inequality; or the forms of pollution, noise, water and food shortages; and other issues and inequalities that are somewhat unique to urban environments [55]. While urban planning is seen as predominantly two-dimensional, works for/with the community and is strongly linked to the government, urban design is focuses on the three-dimensional, total urban form declared 'artistic,' and urbanism links urban form and function, thus linking the two and three dimensions of urban development.

No matter how cities and/or the roles of their urban professions are defined, it is a common subject for discussion, sometimes even raising questions with respect to whether it should exist at all, as some forecast the endgame of the planning system and its ideals that founded the planning movement [63]. There is, in fact, a great deal of uncertainty and confusion over what this role is and what it is likely to be in the future. The perceived role of the planner remains in a state of flux [64]. The 'broader civil society consensus around the need for planning has fragmented, and many people are simply unclear about what the system is for' [65] (p. 23). Additionally, a lack of transparency extends the gap between the planners and the planned, as well as between different forms, sectors, spatial scales or types of 'planner' [66], resulting in mistrust in the planning system, the decisions it produces and the motivations of its central actors [67]. In the transition from the "narrower" world of spatial thinking to the broader world of process and policy formulation, the work of planners became more abstract, less approachable and, ironically, more distant from public expectations about what a plan is or what planners can do. Planning is also often accused of being reactive to real estate or political interests [68]. Subsequently, the serial impacts of

pluralism, liberalism, globalization, risk and rights-based claims have acted in combination to erode an already weak trust in planning and planners [69].

Even more, 25 years ago, Koolhaas already argued that urbanism is dead: 'In spite of its early promise, urbanism has been unable to invent and implement at the scale demanded by its apocalyptic demographics. How to explain the paradox that urbanism has disappeared when urbanization everywhere-after decades of constant acceleration-is on its way to establish a definitive global "triumph" of the urban condition? Now we are left with a world without urbanism, only architecture. The neatness of architecture is its seduction; it exploits and exhausts the potentials that can be generated finally only by urbanism, and that only the specific imagination of urbanism can invent and renew. The death of urbanism-our refuge in the parasitic security of architecture-creates an immanent disaster: more and more, substance is grafted on starving roots. Redefined, urbanism will not only, or mostly, be a profession, but a way of thinking, an ideology: to accept what exists. We were making sandcastles. Now we swim in the sea that swept them away. To survive, urbanism will have to imagine a new newness. What if we simply declare that there is no crisis-redefine our relationship with the city not as its makers but as its mere subjects, as its supporters? More than ever, the city is all we have' [70].

Given these views on the eroded expressiveness, role and even power of the urban professional, how then can they operate and respond with any form of confidence when confronted with the changes currently underway? When entering the age of uncertainty, this calls for a redefinition, maybe rehabilitation, of the urban profession(s).

*2.5. Global Problems*

The urbanized world is and will be confronted with an accelerated and complex set of problems at the global level, which will impact the urban environment and society in every location. The main issues with which the life in cities will experience are in the fields of climate, biodiversity, health and equity.

2.5.1. Climate Change

The following has been said about climate change: 'The scale of recent changes across the climate system are unprecedented and human-induced climate change is already affecting many weather and climate extremes in every region across the globe. Global surface temperature will continue to increase until at least the mid-century, global warming of 1.5 °C and 2 °C will be exceeded during the 21st century. Many changes in the climate system become larger in direct relation to increasing global warming. They include increases in the frequency and intensity of hot extremes, marine heatwaves, and heavy precipitation, agricultural and ecological droughts in some regions, and proportion of intense tropical cyclones, as well as reductions in Arctic Sea ice, snow cover and permafrost. Many changes due to past and future greenhouse gas emissions are irreversible for centuries to millennia, especially changes in the ocean, ice sheets and global sea level' [24].

A concrete manifestation of this is the potential break down of an ice shelf at the bottom of the Thwaites glacier within five years. This could result in sliding of the glacier into the ocean, potentially accelerating sea level rise up to 0.5 m [71–74], which could take a while to effectuate but is irreversible. In the meantime, damage resulting from natural disasters (which are partly caused by climate change) continues to grow and was, in 2021, 24% higher than the year before [75]. The total damage worldwide was EUR 221 billion, of which less than half was/could be insured.

2.5.2. Biodiversity

Climate change impacts and biodiversity loss are two of the most important challenges and risks for human societies. Climate change exacerbates risks to biodiversity and natural and managed habitats. At the same time, natural and managed ecosystems and their biodiversity play a key role in the fluxes of greenhouse gases, as well as in supporting climate adaptation. Nature's contributions to attenuating climate change, partly provided

by the underpinning biodiversity, are at risk from ecosystem degradation resulting from progressive climate change and human activities. In fact, ecosystem degradation through land-use changes and other impacts on natural carbon stocks and sequestration is a major contributor to cumulative carbon-emissions and, therefore, an additional driver of climate change. Therefore, treating climate, biodiversity and human society as coupled systems is a key to successful outcomes from policy interventions, and transformative change in the governance of socio-ecological systems can help create climate and biodiversity resilient development pathways [76].

### 2.5.3. Health

In 2021, the world is overwhelmed by an ongoing global health crisis, which has made little progress to protect its population from the simultaneously aggravated health impacts of climate change. Climate-sensitive infectious diseases are of increasing global concern, and the environmental suitability for the transmission of all infectious diseases is increasing. High temperatures resulted in extreme heat-related health impacts, affecting the emotional and physical wellbeing of populations around the world and resulted in more frequent extreme weather events and increased wildfire exposure; moreover, it has profound effects on food and water security. Measures to curb emissions have been grossly inadequate, contributing to millions of deaths [77,78]. Climate action aligned with Paris Agreement targets could save millions of lives due to improvements in air quality, diet and physical activity, among other benefits [79]. The health co-benefits from climate change actions offer strong arguments for transformative change and can be gained across many sectors, including energy generation, transport, food and agriculture, housing and buildings, industry and waste management [80,81]. Many of the same actions that reduce greenhouse gas emissions also improve air quality [82], while other measures—such as facilitating walking and cycling—improve health through increased physical activity, resulting in reductions in respiratory diseases, cardiovascular diseases, some cancers, diabetes and obesity [83], and urban green spaces facilitate climate mitigation and adaptation while also offering health co-benefits, such as reduced exposure to air pollution, local cooling effects, stress relief and increased recreational space for social interaction and physical activity [84,85]. A shift to more nutritious plant-based diets [86] could reduce global emissions significantly; ensure a more resilient food system; and avoid up to 5.1 million diet-related deaths a year by 2050 [87].

### 2.5.4. Equity

The New Urban Agenda [88] incorporates a new recognition of the correlation between good urbanization and development. It underlines the linkages between good urbanization and job creation, livelihood opportunities and improved quality of life. This implies a vision of cities for all, referring to the equal use and enjoyment of cities and human settlements; seeking to promote inclusivity and ensure that all inhabitants of present and future generations without discrimination of any kind; inhabiting and producing just, safe, healthy, accessible, affordable, resilient and sustainable cities and human settlements to foster prosperity and quality of life for all. The New Urban Agenda is guided by the following interlinked principles:

a.    Leave no one behind by ending poverty in all its forms and dimensions;
b.    Ensure sustainable and inclusive urban economies;
c.    Ensure environmental sustainability.

### 2.6. Urgency

In conclusion, the signs are clear, climate change, biodiversity loss, health impact and increasing inequality show that the urgency to create a fair and livable planet is high. This century will be the era of saving the planet's climate [89]. Will governments act to stop this disaster from becoming worse [90]? The need is high, and the pace at which changes occur only accelerates. In the next 100 years, we will experience many changes

that have happened in the last 1000 years [91], implying that processes of change take place ten-fold faster on average. Warnings and making plans for upcoming risks and change are often blown into the wind. Illustrative in this context is the work of Johan van Veen, who in the decades prior to the last major flood in The Netherlands (1953) warned and, based on scientific investigations, predicted that the country will face a severe risk and experience a major flood. His advice was trivialized, and he was counteracted upon until the 1 February 1953, three days after he presented his plans for the Deltaworks to the Minister of Transport, Public Works and Water Management [92,93]. This points at the need for making advanced plans. Indeed, there is a need for raising awareness and to conceive a new story, a story that is not afraid of envisioning a new landscape [94]. The question, however, is whether this urgency is sincerely felt and whether we will value novel thinking and create the anticipative plans that are needed to turn the serious problems of the future to our advantage?

*2.7. National Confrontations*

These global problems impact every region in a different manner and are of great concern. In The Netherlands, for instance, global problems impact energy, food, biodiversity, climate adaptation and housing agendas:

- The pace of carbon emission reduction needs to be doubled to reach the reduction target for 2030 in order to reduce emissions by 49% compared to the level of 1990 [95]. The reductions realized by end-users, renewable heat and fuel and the reduction in energy demand are all behind schedule. The energy transition, although on its way, lacks the impact required.
- The Dutch food system in its current form is not maintainable. Food is essential for life, but the way it is produced causes environmental and climate problems. The amount of land and resources used threaten biodiversity and is unhealthy, and overconsumption of food increases obesity and other food-related diseases [96].
- The deposition of nitrogen has profound impacts on biodiversity and the quality of nature and prevents building activities. Not everything is possible anywhere [97]. Moreover, there are stealthy effects of nitrogen on nature and health [98], and reducing deposition is surrounded with doubts about its execution [99].
- The country needs to adapt to the local impacts of climate change, such as fast change in climatic conditions; effects on sea level rise; larger differences between high and low water levels in rivers; and increased probability of droughts. Specific weather conditions occur over prolonged periods (dry, hot, cold and wet), and urban areas will face extreme precipitation events, heat and rainfall [100].
- The country has a shortage of affordable and quality of housing. In the period 2021–2034, the total amount of households increases with 849.000, a growth rate of 10.5% [101]. Moreover, the Delta Commissioner has announced that a large part of the current housing development areas is in highly risky areas, and it can be questioned whether these houses should be built in these areas and, if so, with what fundamental adaptations [102].

All these problems urge for integrated planning, based on the water, soil and ecological systems. As space is scarce, this requires innovative spatial concepts in which different land uses need to be combined; sectoral approaches are abandoned; and a mix of land use is unavoidable [103]. Several attempts to unify large questions in coherent plans for the country have been presented. NL2120 proposes a long-term vision for The Netherlands, taking ecological and water systems as the point of departure [104]. NatureRich Netherlands proposes to transform the country to reserve 50% of its area for nature in a budget-neutral manner while simultaneously solving the agricultural transition, overcoming the nitrogen problem and realizing enough attractive and affordable housing [105]. NL2121-A land with a plan [106] advocates for a research-by-design approach to find solutions for the battle for space in the century of big transitions [91].

In conclusion, on the one hand, the role of urban planning, design and urbanism is under permanent debate, is scrutinized and is often misunderstood or undervalued by the public; on the other side, the problems that affect life in cities are increasingly complex and potentially harmful. This tension asks for a rethink of the work, role and competencies of the urban professional.

## 3. Methodology

In order to investigate (Figure 1) what a future urban professional may be confronted with and how to prepare for these changes, a series of in-depth interviews of 60–90 min each has been conducted. Twenty-five thought leaders were selected carefully because they work in different sectors of the industry; are from different disciplinary backgrounds; and are spread over geographical regions and levels of scale. Amongst the interviewees were people from (semi-)government; policy and politics; the building and construction sector; private sector urban design and architectural firms; academia; and NGOs. Four interviews were undertaken with people from abroad (Australia, UK and US); twelve with people from The Netherlands, of which several work internationally; and nine with people from the North-Holland region. This provided a spread over disciplines and background knowledge, focus and perspective.

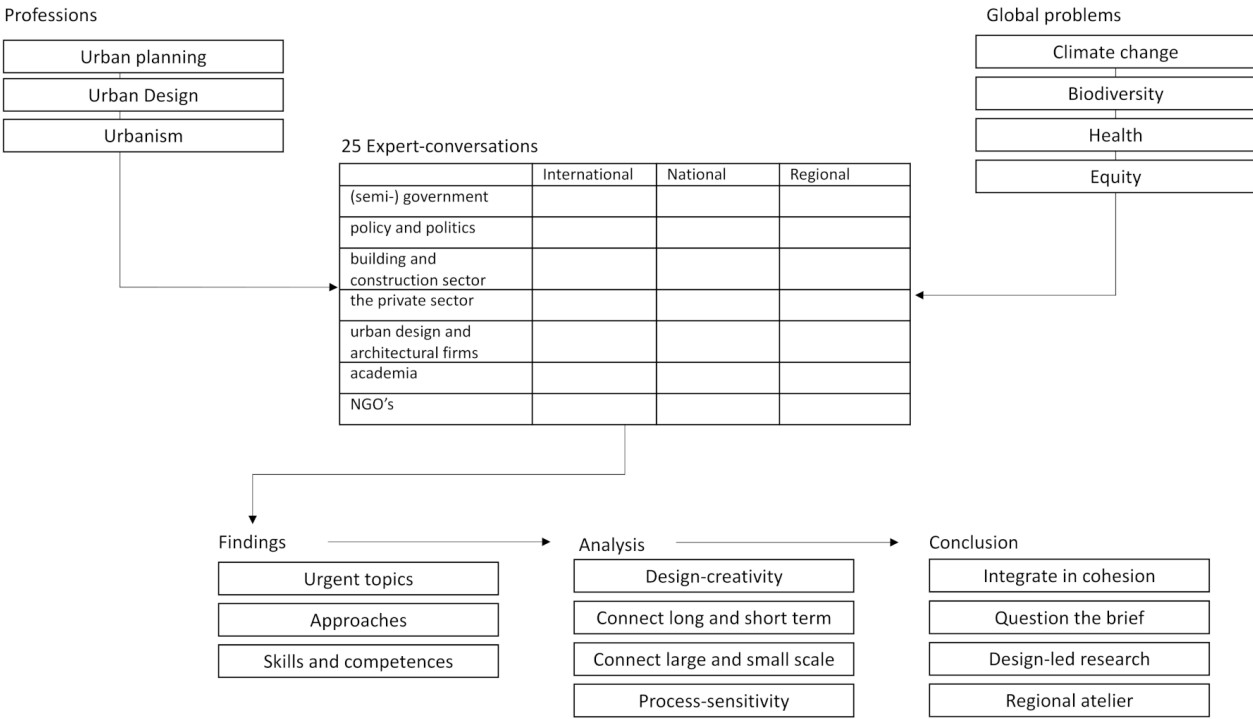

**Figure 1.** Methodology.

The interviews were semi-structured along two main questions: what are seen as the most urgent items/topics of our time, and what are the implications of this for the education of the current generation of future urban professionals? The interviews were set up as a conversation rather than a questionnaire.

After the interviews, the conversations were transcribed in the form of an essay, which are bundled in a separate publication [94]. From each interview, the main points have been collected and analyzed along three fields of interest: urgent topics; suggested conceptual approaches; and required skills and competences. The result of this analysis is reflected in the Findings section.

## 4. Results

The transcribed results derived from the interviews have been summarized and reproduced in the form of a short story. The content is analyzed and structured in three coherent parts: urgent topics, suggested approaches and required skills and competences.

### 4.1. Urgent and Upcoming

The major urgent topics mentioned in the conversation can be subdivided into five categories:

- First and foremost, biodiversity loss and climate emergencies are expected to dominate challenges for urban professionals. This has profound impacts on the role for nature-based design, adaptation of land use to climatic impacts and adjustments to the water system.
- Secondly, societal polarization is observed as an increasing problem, resulting in distrust and fear amongst citizens, who on their turn show increased opposition to plans and have sharp, confronting opinions. This is exaggerated by the manner social media is influencing political decision making and the debate in general. The distribution of wealth is under pressure as, for instance, the battle for housing illustrates, placing equity issues and just urbanism on the agenda.
- A third cluster of topics is found around circularity, including reuse and recycling of materials; use of prefab and wooden materials; the energy transition and reduction in carbon emissions; and smart use of resources. Hyperlocalization of urban flows, for instance, in the manner food is grown with nature-inclusive and urban agriculture methods is essential for closing cycles.
- A fundamental point is made regarding economic mechanisms. If a transformation of urban environments is desired and even deemed necessary for human survival, the current economic laws must be put to use for supporting that transformation. Principles such as degrowth [107,108], finance of sustainable building materials, the affordability of the housing stock and novel international patterns of job locations require an alternative view, i.e., Modern Monetary Theory [109]. Current political contrapositions and interests, generally reaffirming the existing financial–economic reality, need to be overcome.
- The fifth category of urgencies is related to urbanization. In a country such as The Netherlands, the pressure on land is high, land is scarce and the battle for space is ongoing. At the same time, not every area is free of the risk of climate impacts, which in the end increases pressure. Additionally, the growing housing demand is translated in higher density urban centers while urban systems themselves are no longer seen as stable as they were, resulting in social unrest perverse mechanisms related to housing prices and a danger of reduced quality of living. Simultaneously, rural areas are left to their own devices, ultimately ending in a conservative worldview where livability is under pressure and amenities are reduced to minimal levels. Finally, there is a well understood need for green urbanism, bringing nature as a driving force in urban development and viewing nature as an inseparable part of the urban environment.

By contemplating the range of urgencies, it is impossible to solve all problems and changes one by one. It is essential to see problems in conjunction with each other and to learn from nature in order to be able to deal with unprecedented changes and extremes. This is even more needed given the complexities of spatial implications of the range of (urban) problems, the scope of time horizons and the breadth of involved people and organizations. The question is whether we are prepared for this new era in which uncertainty is the new norm. We will not succeed if we try to deal with this new paradigm in the manner that we solved the problems of the old one. One way or another, it is necessary to escape 'currently adopted' policies. From the conversations, a striking tension arises. Two opposing worldviews emerge, realism vs. idealism, which face difficulties to strengthen each other by using one another's perspective. It results in persistence about the positions taken and further polarizing and standing with their backs to each other (Figure 2).

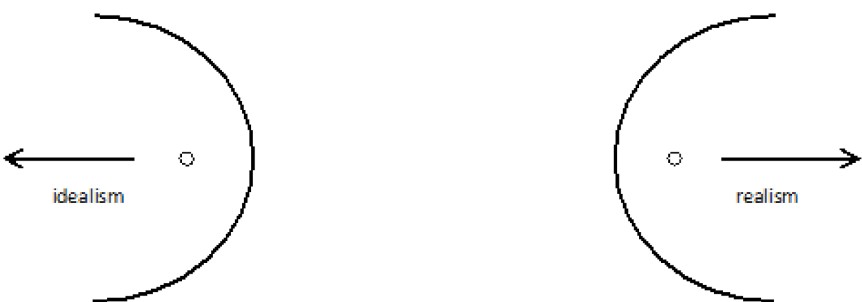

**Figure 2.** Tension between the focus on realism vs. idealism [94].

On the one hand side, the realists place people as central and allow nature whenever possible. Thinking is dominated by financial–economic laws, and they strive for robustness and inertia of the system. In this manner, the need for slowness is fed. On the basis of technological knowledge and linear standards and norms, certainties are sought in political, policy and participation processes. Opposite of this view, idealists see man as part of nature. This natural system is at risk due to climate change and loss of biodiversity. The search for landscape-driven sustainability implies transitions with respect to the manner we grow our food and generate energy. When parallel resilience is increased, dealing with uncertainties becomes much easier. The threat of degradation of the natural systems urges us to act swiftly.

Both perspectives have their value. People need certainties in their lives and a continuation of current living conditions, while their survival depends at the same time on the resilience of their environment, especially when uncertainties rise. Therefore, it is essential that both perspectives are united.

Luckily, we stand at the brink of transitioning to a new mental model in which the siloed manner of thinking is replaced by relational thinking and oriented on the long term (Figure 3). To complete this transition, innovators and early adopters (together, pprox... 14–20% of the population) must embrace the new set of values. Currently, this group is ca. 15% large [110]; hence, we are about to shift to the new paradigm. Therefore, it is no surprise that the current period is chaotic and disruptive, full of innovation, emerging potentials and experimentation. It is time to learn to embrace this chaos as it is part of the transition we currently undergo [91].

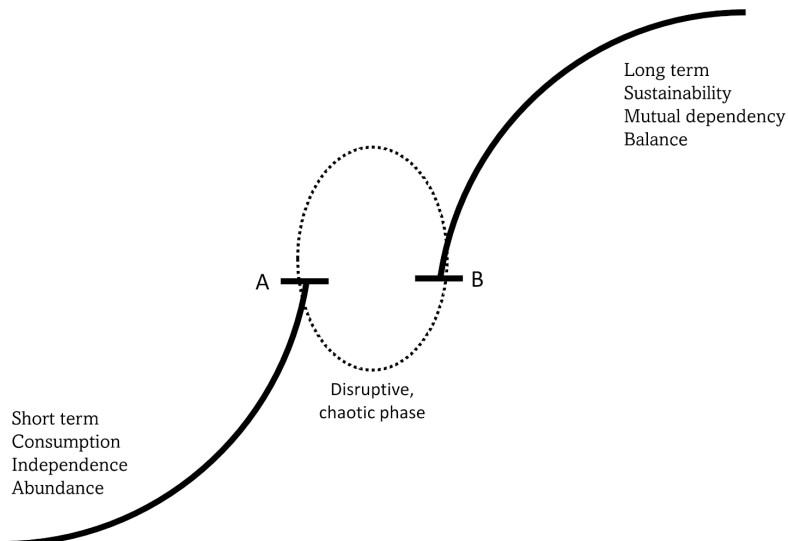

**Figure 3.** Standing at the brink of a new time (B), while currently being in a period of disruption (A) (source: [94,111–113].

This requires that we embrace the new paradigm and adopt a new methods of thinking. Sustainability on Earth will be strongly determined by the way we think and act with a long-term perspective with respect to being a good ancestor [114]. The expertise of thinking in robust systems must be integrated in an idealistic perspective. The permanence of the natural system benefits from a solid financial foundation for political decision making and policy. On the other hand, future thinking and incorporating the impact of actions here and now for future generations shall play a larger role in determining financial-economic frameworks by becoming more creative and design-oriented. This is the only way to overcome soloistic solutions. First, an integrated view must define the task at hand by interlinking designed creativity and process sensitivity, connecting long- and short-term periods as well as large and small scales (Figure 4). In this manner, the solution will not be a desired blueprint of the future but consists of an adaptive task, process and vision.

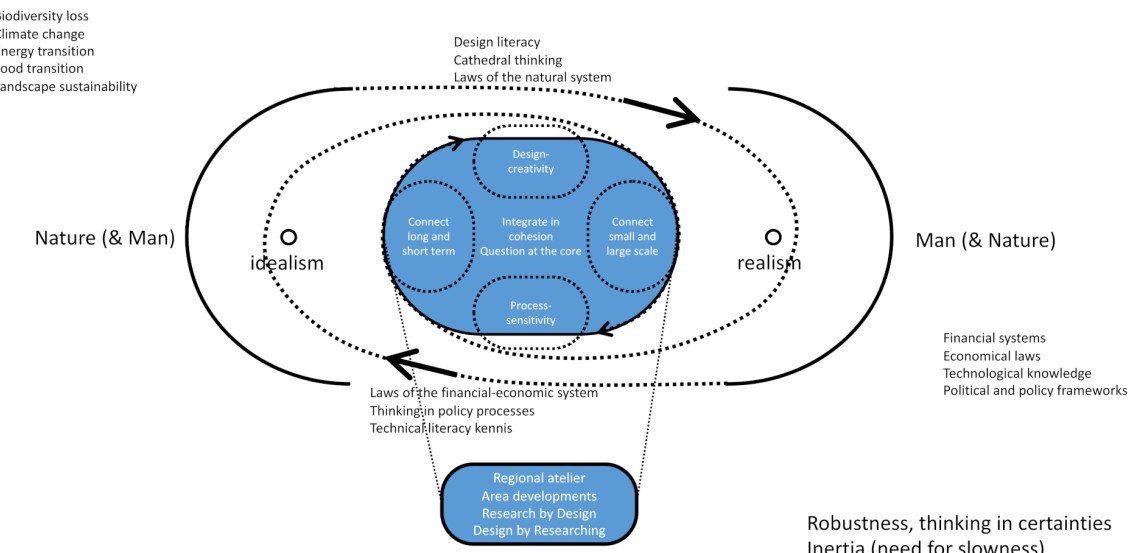

**Figure 4.** Reconnecting urgency and robustness [94].

*4.2. Suggested Approaches*

The conversations also shed light on the types of approaches that will be useable in dealing with future urban problems:

- What if questions.

    To respond to fundamentally different futures, climate change impacts will need to be taken as the core questions of research and education. Muddling through is not an option. In this context, confronting planning for urban environments with 'what-if' questions is suggested. In apocalyptic design studios, unorthodox and effective solutions embracing rigorously different futures can be explored and imaginary scenarios can be crash-tested to anticipate disruptive futures. Challenging questions are posed; curiosity and improvisation are stimulated; and the headspace for unexpected outcomes is created, digging into the so-called unknown unknowns. In the process of raising awareness for large transitions, a strategy of 'wait-and-plan' could be applied in which the processes of change become necessities, the real urgencies become clear and the need for innovation and 'away from the average' [115] planning is created. This process emphasizes fluidity, creativity and temporality and raises tensions and uncertainties such that adaptive, agile and resilient designs can be conceived, anticipating black swans [116] and nonlinearity. In this environment, innovative thinkers, who deal with change through creativity and intuition, phantasy and imagination, flourish and will come up with contingencies in designs, incorporating space for

something it is not meant for, which is exactly the type of flexibility in planning that is needed in an era of uncertainty.

- Teaching a region.
  The second suggestion is to start teaching the region instead of a series of subjects. This could take a shape in the form of a regional design atelier in which spatial solutions, scales and time horizons are integrated in a design-led method of approach. When the region is the subject of spatial investigations, the findings can be stacked to draw conclusions based on multiple years of aligned research. This demands a structured process, starting with the definition of a brief, spatial-analytical research and data visualization, followed by a 'research-by-design' phase, in which the power and limitations of a range of spatial interventions are determined, after which the design propositions are sketched, modelled and reviewed. The subject of study, the region, is viewed as an organism that is constantly undergoing change and transforming and seeking optimal resilience. This requires a permanent process of questioning the brief at multiple scales, which includes constantly asking the following question: 'What will the adaptability in the future be?' and how can the plan be beneficial in multiple ways. By taking the region as the focus of teaching, but also for policy making and social cohesion, it is deemed to become an accepted part of the planning process and political decision making.

- Non-rationality.
  A third element arising from conversations is the new mentality that is essential for dealing with the complexity of changes. Creativity, intuition and imagination for a new set of values aim to contribute value instead of depletion. In this context, we can learn from indigenous ways of knowing, i.e., oral learning. When we make use of our senses, beyond rationality, we can connect to the 'Land,' to Country and learn from the stories of the traditional owners of the land. In the Dutch context, this would mean that we start collecting stories of traditional knowledge, the way the land was used and treated and the cultural dimensions of living together in the lowlands of the delta. This understanding can be used when decisions are made for the future and how these fit (or misfit) with the regional cultural system. This implies a diversion from the belief in technology and hyper-specialization as the one and only method to deal with problems (Table 1). Design can soon be freed from technology and return to its purpose of creating a living condition that makes people happy.

- Integrated design.
  In times of growing uncertainties, the need for a beckoning perspective that is coherent and attractive is needed more than ever. Instead of separating problems in sectoral departments and sections, integration in a design is unavoidable for shaping this perspective. In a national area development process, guided and led by young people, such a perspective solves the tension between the long period of time needed for urbanization processes and the urgency of problems, which require immediate action. In a national lab for virtual collaboration, long-term and large-scale perspectives should be synergized in a holistic vision connecting understanding, awareness and enthusiasm. Integrated thinking and planning would connect content and process; public and private interest; and place participation and communication at the heart of the process to gain nationwide support. In such a national area, development thinking should take ecological and water systems as the point of departure for an integral plan starting with the landscape and nature, in which land is reversely engineered and de-cultivated. This prioritized process should be realized and exploited under the responsibility of the government. This would then be followed by housing and be exploited by the market. In this manner, a strong link can be established between the values of water, ecology and soil systems in combination with the economic value of the land, and synergies emerge between nature and housing. The role of the

(national) government is to be a fair and strong entity, guiding content and process with overview.

-   Landscape-driven design.
    The final cluster of suggested approaches to urban and regional planning is to take the landscape as the first point of entrance. A landscape-driven design and planning process would take the understanding of the landscape, its water and ecological systems, as well as its cultural heritage and beauty, as a guiding principle in any planning, design, development and building process. When landscape and nature is placed first, second and last when designing our (urban) environment, natural principles, processes and concepts will direct urban patterns, uses and functionalities, respectively. The 'ecologized' city will be turned into a rewilded bioregion by applying 'building with nature principles.' For land-use, coastal protection, the food system, life and learning from nature will make society healthier, more resilient and happier. Green, nature- and landscape-inclusive urbanism places (socio-ecological) resilience, climate sensitivity, adaptivity and flexibility at the core of urban development.

**Table 1.** Shift from mechanical to organic worldview.

| From: Mechanical | To: Organic |
|---|---|
| Problem solving | Organic self-organization and emergence |
| Contract | Complexity |
| Linear | Non-linear |
| 'Singlicity' | Multiplicity |
| Controlling the water system | Dynamic water system shapes the landscape |
| A certain world with patches of uncertainty | An uncertain world with patches of certainty |
| Homogeneity, regulatory, globalized | Localized expertise, methods, and culture |

*4.3. Required Skills and Competences*

Topics such as urgent problems, changes and transformations and how to approach the era of uncertainty also bring novel ways of thinking, competences and skills and roles for the future urban professional to the foreground (Table 2). The future urban professional needs to be able to think out of the box and develop visionary plans and concepts. Secondly, a high level of process sensitivity is required in order to deal with complexities and polarizations in society and politics. Finally, actual knowledge is required in fields that will determine the success in planning for an uncertain future: climate literacy, modern monetary theory and (resilient and regenerative) systems thinking to name a few.

**Table 2.** Desired skills and roles for the future urban professional.

| | Way of Thinking | Competences/Skills | Roles |
|---|---|---|---|
| Visionary view | Oral learning<br>Long-term thinking<br>Multiple scale<br>Intergenerational<br>Out of the box<br>Intuitive thinking<br>Adaptivity, survivability, preservability<br>Frugal and bright future<br>Connecting spatial patterns with technical attributes<br>City/landscape as an organism | Design skills<br>Dare to innovate<br>Anticipate fast changes and surprise<br>Ability to improvise<br>Creativity<br>Take initiative<br>Embrace complexity<br>Conceptual analytical<br>Visualization | Poetic designer, beautification idealist<br>Futurist<br>Design team builder, bring multiple disciplines together |

**Table 2.** *Cont.*

|  | Way of Thinking | Competences/Skills | Roles |
|---|---|---|---|
| Process sensitivity | Inter- and cross-disciplinary Building communities Social cohesion, citizen engagement Transparent and debatable design process | Collaborative problem solving Teamwork and trust Network building Agility Feeling for society Reflective empathy Negotiation Comprehend the policy process and decision making Understand/convince opposing powers and polarization | Professional citizen, connect professional practice with citizens Participation artist Take a leadership role Organizer of community support for decisions Designer of the process Builder of social capital and enhance learning capacity Creator of environment for self-determination |
| Actual knowledge | Specialist generalist Place expertise in broader context Resilience and regeneration; socio-ecological systems De-cultivation of the landscape Degrowth Financial systems for sustainability | Climate literacy Economic mechanisms Value creation Urban systems logistics Curiosity Data interpretation | Sharp and Friendly Advisor Curator of the city Presenting complex information in attractive and easy understandable way |

## 5. Discussion

In this article, the contours of the urgencies, ways of thinking and competencies of a new urban profession(al) in an era of uncertainty have been collected and aligned based on 25 intensive conversations with internationally renowned experts. Despite the depth of information and thought, 25 people can never represent the entire urban professional community. Therefore, the conversations must be seen as a gigantic first step that is utterly small at the same time.

In the conversations, an abundance of well-evidenced points has been brought to the foreground. However, the complexity and expectations of tasks, roles and competences of future urban professionals are almost too large to distill a description of the urban profession. Given the range of subjects, it is next to impossible to be an urban professional that suits everyone's needs and desires.

In this article, professions are limited to the core urban planning and design disciplines. It would be enriching to broaden the scope of disciplines, for instance, with landscape architecture, urban development/area development, social and just urban analytics, urban economics or urban environmental systems.

Further research can be valuable with respect to the following aspects:

1.  To extend the scope and number of future talks beyond the current and grow to 50, to 100 and to 500 in the next 5 years;
2.  To select priorities in the abundance of requirements, expectations, competencies, outcomes and knowledge that the new urban professional should comprehend. This should be based on further research into the most important transformational changes.
3.  To elaborate a process design that can function as the core policy process at national and regional levels. Due to the fact that every area and local context differs, it is a mistake to define a prescriptive set of competences. However, it seems clear that, in a shift from a technocratic method of responding to problems to greater empathy,

an increased 'feeling for society' and increased oral ways of knowing are needed to discover the DNA of the region and turn that into a beckoning perspective.

## 6. Conclusions

We enter a novel era that will be dominated by larger uncertainties. This implies that existing practice in the fields of urban planning, urban design and urbanism no longer suffices. No matter what is traditionally observed as the role for the planning professional, as discussed in this article in the current time period, the professional cannot count on much support from society. Partly, this is due to the fact it has become more and more unclear what the role is and what the professional influence is on actual urban developments. Moreover, the complexities of tasks, data and finding a concise brief have become difficult to comprehend from a certain profession. This calls for cohesion and collaboration between disciplines and a joint search for an adaptable urban future.

This means that there is a need for a renewed position of the urban professional of whom he or she is expected to be a specialist generalist who can think outside the box; develop inspiring well-evidenced visions; and gain support from citizens and politicians alike. In this article, 25 conversations point at a new urban professional who can integrate and collaborate; is utterly creative and agile; understands the threats of the near future; and can formulate and present a convincing answer to the challenges using all senses by applying non-rational competencies.

The discussed climate emergency and other rapid changes that are difficult to predict make it necessary to be ready for the crash. Therefore, we need to conduct some crash-testing of design propositions that work in extreme, maybe catastrophic climate conditions, preventing the apocalypse from becoming a reality. Such an approach is observed as landscape-driven because the resilience of landscape and nature can direct us towards ways to plan for our (human) environments that are at their highest adaptive capacity possible. This requires coherent thinking in interdisciplinary ways and in regional and national area-oriented development processes. Integrated landscape design, in which problems, solutions and opportunities are synergized and unified, need to be utilized with no exceptions.

**Author Contributions:** Conceptualization, R.R. and R.C.; methodology, R.R.; validation, R.R. and R.C.; formal analysis, R.R.; investigation, R.R. and R.C.; data curation, R.R.; writing—original draft preparation, R.R.; writing—review and editing, R.C.; visualization, R.R. All authors have read and agreed to the published version of the manuscript.

**Funding:** This research received no external funding.

**Institutional Review Board Statement:** Not applicable.

**Informed Consent Statement:** Not applicable.

**Data Availability Statement:** Not applicable.

**Conflicts of Interest:** The authors declare no conflict of interest.

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
