# Peer review of "The New Urban Profession: Entering the Age of Uncertainty"

_urbansci, doi:10.3390/urbansci6010010_

Round 1

Reviewer 1 Report

  1. The authors should ask the help of native English speaking proof reader,
    because there are some typo and linguistic mistakes that should be fixed.
  2. It is recommended that the article be enhanced by a flowchart explaining the research technique.
  3. It is advised to organize Discussion and Conclusion part much properly with additional explanation/details.

Reviewer 2 Report

Dear authors, an interesting problem.

However, there is no logical connection: how will a change in thinking, new competencies have an impact on urban design, urban planning, and the economy of the city?

The mental model in question, relational thinking

they should promote the adoption of a new paradigm of city values.

The article lacks tools for assessing the involvement of city residents in the processes of changing the urban landscape.

The results of the conducted research in the number of respondents surveyed are not very presentable.

Reviewer 3 Report

The authors point to important issues. The paper reminds me of Keynes's quote 

“The master-economist must possess a rare combination of gifts .... He must be mathematician, historian, statesman, philosopher—in some degree. He must understand symbols and speak in words. He must contemplate the particular, in terms of the general, and touch abstract and concrete in the same flight of thought. He must study the present in the light of the past for the purposes of the future. No part of man's nature or his institutions must be entirely outside his regard. He must be purposeful and disinterested in a simultaneous mood, as aloof and incorruptible as an artist, yet sometimes as near to earth as a politician.”
― John Maynard Keynes

The paper is carefully written. i only found one typo. In the paragraph below, suffice should be suffices. 

We enter a novel era that will be dominated by larger uncertainties. This implies that existing practice in the fields of urban planning, urban design and urbanism no longer suffice. No matter what traditionally has been seen as the role for the planning professional being an artist, a regulator, or a policymaker today the profession cannot count on much support from society. Partly this is due to the fact it has become more and more unclear what its role actually is and what the professional influence is on the actual urban development.

However, I think this paragraph could be written much more clearly. I suggest that the authors reread the manuscript and find additional instances where the language could be streamlined.

Also consider this paragraph below. "predict nor projected" lacks parallelism and confuses tenses. Better to write "predict or project." Better still i think would be simply "predict."

Also ways is plural. so is should be are.

The discussed climate emergency and other rapid changes that are difficult to predict nor projected, make it necessary to be ready for the crash. Therefore, we’d better have done some crash-testing of design propositions that work in extreme, maybe catastrophic climate conditions, preventing the apocalypse from becoming a reality. Such an approach is seen as landscape driven, because the resilience of landscape and nature can direct us towards ways to plan for our (human) environment that is at its highest adaptive capacity possible. This requires coherent thinking, in interdisciplinary ways and in regional and national area-oriented development processes. Integrated landscape design, in which problems, solutions and opportunities are synergized and unified need to be usance, no  exception.

I suggest checking through the paper and eliminating unnecessary words. (I am a fan of Deidre McCloskey's book "Economical Writing."). 

Round 2

Reviewer 2 Report

Good afternoon,

thank you for the opportunity to act as a reviewer.

Your author's scientific position is quite original and interesting.